# From Spice to Survival: The Emerging Role of Curcumin in Cancer Immunotherapy

**DOI:** 10.3390/cancers17152491

**Published:** 2025-07-28

**Authors:** Jacob M. Parker, Lei Zhao, Trenton G. Mayberry, Braydon C. Cowan, Mark R. Wakefield, Yujiang Fang

**Affiliations:** 1Department of Microbiology, Immunology & Pathology, Des Moines University, West Des Moines, IA 50266, USA; 2The Department of Respiratory Medicine, the 2nd People’s Hospital of Hefei and Hefei Hospital Affiliated to Anhui Medical University, Hefei 230011, China; 3Department of Surgery, University of Missouri School of Medicine, Columbia, MO 65212, USAwakefieldmr@health.missouri.edu (M.R.W.); 4Ellis Fischel Cancer Center, University of Missouri School of Medicine, Columbia, MO 65212, USA

**Keywords:** curcumin, immunotherapy, tumor microenvironment, adjuvant, immune checkpoint inhibitors

## Abstract

Curcumin is a widely available spice found in turmeric that may have anti-cancer effects. Additionally, drugs that leverage the immune system to eliminate cancer cells, called Immunotherapeutics, have revolutionized how many cancers are treated. The properties of curcumin align with the goals of immunotherapy; therefore, this review examines how curcumin can be utilized in conjunction with immunotherapies to treat cancer. We examine how it can alter the tumor microenvironment and how it will impact immune cells, and finally, we explore how different immunotherapies may be affected by curcumin. We suggest that curcumin may be beneficial in conjunction with immunotherapies in certain contexts, but it also faces clinical challenges such as bioavailability.

## 1. Introduction

Cancer is one of the leading causes of death globally, with 2 million diagnoses annually [1]. Despite advances in surgery, chemotherapy, and radiotherapy, these modalities often fall short, resulting in resistance, recurrence, and severe side effects. There is a pressing need for new therapeutics that minimize the shortcomings of traditional therapies. Immunotherapies such as immune checkpoint inhibitors (ICIs), cancer vaccines, and adoptive cell therapies have done this and revolutionized the landscape of cancer therapeutics by offering improved side effects and lasting responses in many patients [2]. These therapies aim to “wake up” the immune system and help it recognize and destroy cancer cells by targeting signals that inhibit immune cells.

However, immunotherapies are far from perfect, with response rates plateauing around 20–40% [3]. The efficacy of these drugs can be hampered by the immunosuppressive TME, which limits the function and number of immune cells necessary for a strong response in immunotherapy. To address this, extensive research has gone into finding novel ways to modulate the TME and immune system to promote stronger anti-cancer responses [2]. Curcumin, a polyphenolic phytochemical, has been shown to modulate the TME and immune cells in many interesting ways. Making curcumin a potential adjuvant in combination with immunotherapies [4].

Curcumin is derived from the spice turmeric, and it has been used as a medicinal compound dating back over 5000 years [5]. Today, curcumin is studied as an anti-inflammatory and anti-cancer compound because of its ability to modify key biochemical pathways including nuclear factor kappa-light-chain-enhancer of activated B cells (NF-*κ*B), Janus-activated kinase (JAK)/signal transducer and activator of transcription (STAT), cyclooxygenase-2 (COX-2), nuclear factor erythroid 2-related factor 2 (Nrf2), caspase cascades, vascular endothelial growth factor (VEGF), and matrix metalloproteinases (MMPs) [6,7,8,9,10,11]. By modulating these pathways, curcumin can impact immune cells, including T cells, macrophages, dendritic cells (DCs), and natural killer (NK) cells [12,13,14,15]. By exerting these effects, curcumin targets multiple cancer hallmarks and may be useful in treating multiple types of cancers.

This review aims to provide a comprehensive overview of how curcumin exerts its anti-cancer and anti-inflammatory properties, with a special focus on how its immunomodulatory effects may allow synergistic effects with immunotherapies. By exploring this topic, we hope to highlight the potential of curcumin and offer new avenues for the treatment of cancer.

## 2. Mechanisms of Action of Curcumin

### 2.1. Anti-Inflammatory Properties

One key feature of curcumin is its anti-inflammatory properties. Curcumin primarily exhibits its anti-inflammatory effects by downregulating key signaling mediators, most notably NF-κB. However, COX-2, mitogen-activated kinases (MAPKs), and Janus kinases are also affected by the presence of curcumin. It also inhibits pro-inflammatory cytokines, including tumor necrosis factor (TNF-⍺), interleukin (IL) 1, IL-2, IL-6, IL-8, IL-12, and IL-5 [8].

Curcumin is a potent NF-κB blocker, a key regulator in the inflammatory response. NF-κB can influence the expression of chemokines, cytokines, adhesion molecules, kinases, and enzymes that promote inflammation [6]. The mechanism by which NF-*κ*B is inhibited starts with curcumin directly interacting with inhibitor of kappa B kinase (IκKβ), which blocks the phosphorylation of IκB. This prevents IκB from binding to NF-κB, which prevents NF-κB from translocating to the nucleus and affecting the transcription of pro-inflammatory genes [16].

While curcumin directly interferes with NF-*κ*B, it can also do this indirectly. When oxidative stress is high, NF-κB signaling is promoted. Since curcumin also functions as an antioxidant, it may decrease oxidative stress and downregulate the NF-κB pathway [17,18]. Curcumin also inhibits the downstream effects of NF-*κ*B, such as IL-1, IL-6, IL-8, COX-2, and nitric oxide synthase (iNOS), which further reduces the inflammatory environment [16,19,20].

Additionally, curcumin modulates the TNF-cytokine, which has pro-inflammatory effects [8]. More research is necessary to understand the mechanism by which curcumin affects TNF-⍺, but it is believed curcumin downregulates NF-*κ*B and STAT3, subsequently decreasing TNF-⍺ production [7]. Curcumin may also disrupt the signaling between TNF⍺ and the TNF receptor, which prevents the pro-inflammatory effects from occurring [21]. Further research to solidify our understanding of curcumin and TNF-⍺ is necessary.

### 2.2. Antioxidant Activity

Curcumin is known for its antioxidant effects, primarily due to its chemical structure. The hydroxy moieties at the 4 and 4′ positions on the phenyl rings allow curcumin to donate hydrogen to and neutralize reactive oxygen species (ROS) [22]. This mechanism can be seen in Figure 1. This ability to scavenge ROS is why curcumin is considered an excellent antioxidant and has been proposed to be particularly effective at protecting the cell membrane from oxidative stress [22,23].

In addition to its direct antioxidant effects, curcumin can influence antioxidant pathways, particularly in the Nrf2/ECH-associated protein-1 (Keap1)/antioxidant response elements (ARE) pathway. Curcumin exhibits keto-enol tautomerization of its β-diketone group, and in the enol form, curcumin can bind to cysteine residues on Keap1 [24]. This causes a conformational change, which results in Nrf2 translocating to the nucleus. Nrf2 translocation is regulated by phosphorylation of its threonine and serine residues by MAPK, phosphatidylinositol 3-kinase (PI3K)/Akt, and protein kinase C (PKC) [9]. This ultimately leads to the expression of ARE genes, which results in the production of heme oxygenase-1, NAD(P)H-quinone oxidoreductase-1, superoxide dismutase, thioredoxin-reductase, glutathione reductase, and catalase, among others [25,26,27,28,29]. Each of these products serves to reduce ROS concentration within cells.

It has been established that curcumin regulates transcription factors such as Nrf2, NF-κB, and STAT3 [7,9]. These pathways have downstream implications on important pathways including PI3K/Akt/mTOR, Ras/Raf/MEK/extracellular signal-related kinase (ERK), GSK-3β, and p53, among others [29]. These pathways are integral to cell survival, proliferation, DNA repair, and apoptosis, but they also have implications for the cell’s proper response to oxidative stress. By modulating these pathways, curcumin not only reduces the amount of oxidative stress but also helps the cell regulate itself properly. Modulating these major pathways highlights curcumin’s potential therapeutic role as well.

### 2.3. Apoptosis Induction

Effective cancer therapeutics must directly or indirectly result in the apoptosis of cancer cells. Apoptosis can occur through intrinsic or extrinsic pathways visualized in Figure 2. Curcumin induces apoptosis in different types of cancer cells by affecting apoptotic mitochondrial pathways or apoptotic death receptors [30]. Curcumin affects major apoptotic pathways, including decreased expression of Bcl-2 and Bcl-XL and increased expression of p53, Bax, Bak, PUMA, Noxa, and Bim [31,32]. By affecting these components, curcumin has been demonstrated to induce apoptosis through both intrinsic and extrinsic pathways.

#### 2.3.1. Curcumin and Intrinsic Apoptosis

Intrinsic apoptosis occurs through internal cell signals such as DNA damage, oxidative stress, or other factors. The intrinsic apoptosis cascade revolves around changes in the mitochondria. A decrease in mitochondrial membrane potential has been demonstrated as a crucial and early step in the apoptosis cascade because this disrupts ATP synthesis and releases cytochrome C. Multiple studies have demonstrated that curcumin can decrease the mitochondrial transmembrane potential, which results in increased apoptosis [30,33,34]. This is the first step in which curcumin influences apoptosis.

Bcl-2 and Bcl-XL are pro-survival signal molecules, and their expression results in the growth and proliferation of cells. Bax is the foil to Bcl-2; it is a pro-apoptotic protein that is activated by proteins such as Bim and PUMA [35]. By downregulating Bcl-2 and upregulating Bax, Bim, and PUMA, curcumin intrinsically induces apoptosis [36]. Studies examining the Bax/Bcl-2 ratio found that as the ratio increased, so did the subsequent release of cytochrome C into the cytosol [37,38]. Free cytosolic cytochrome C combines with Apaf-1 and caspase-9, which forms an apoptosome complex. This subsequently cleaves procaspase-3 into active caspase-3, which results in apoptosis [39]. In addition to affecting the expression of Bcl-2 and Bcl-XL, curcumin also affects their functioning within the cell. NF-*κ*B is associated with increased expression and activity of Bcl-2, Bcl-XL, and apoptosis resistance [40]. Curcumin inhibits NF-*κ*B as previously described, which prevents cell proliferation and promotes apoptosis.

Curcumin can also induce apoptosis by affecting p53 function [41]. In healthy cells, p53 is implicated in DNA repair, cell cycle arrest, and apoptosis. In cancer cells, p53 can become mutated and its functions can be inhibited, resulting in uncontrolled proliferation. Curcumin has been shown to induce p53 DNA binding and increase p53 activity, which promotes the formation of Bax, which is pro-apoptotic [42]. Curcumin also inhibits the activation of Akt, a signal that can interfere with p53 functions, which allows for increased p53 activity [43]. Curcumin may also be able to reduce levels of mutant p53; however, further research is necessary on this topic [44].

The PI3K/Akt/mTOR pathway is also affected by curcumin. This pathway is involved in the growth and proliferation of cells, and one outcome is the release of Bcl-2. Within cancer, this pathway can become overexpressed, which results in quicker cell proliferation. Curcumin can directly interact with Akt, PI3K, and other growth factors, including epidermal growth factor receptor (EGFR) [45,46], but it also indirectly affects this pathway by inhibiting IκKβ, which is an upstream regulator of mTORC1. This results in inhibition of the PI3K/Akt/mTOR pathway, therefore inhibition of cellular growth and promotion of apoptosis [16,47]. Inhibition of the PI3K/Akt pathway promotes apoptosis because Akt can no longer phosphorylate the Bcl-2-associated agonist of cell death (BAD) protein at Ser 136, releasing the pro-survival Bcl-2 molecule [48,49].

#### 2.3.2. Curcumin and Extrinsic Apoptosis

Curcumin has been implicated in upregulating death receptors on cells, including death receptor-4 (DR4) and death receptor-5 (DR5) [50]. In one study, curcumin was found to upregulate the Fas receptor gene in a dose-dependent manner [51]. Additionally, curcumin has demonstrated the ability to increase the Fas ligand (FasL). The combination of these two effects results in increased Fas-associated apoptosis [50]. It has also been shown to potently upregulate DR5 and DR4, which enhances TNF-related apoptosis-inducing ligand (TRAIL) induced apoptosis [52,53]. Once TRAIL binds to the DR, an apoptotic cascade is initiated in which the Fas-associated death domain adaptor protein (FADD) is recruited. This subsequently recruits procaspase-8, and this complex composes the death-inducing signaling complex (DISC). Within this complex, procaspase-8 is converted to the active caspase-8, which goes on to activate other caspases, including caspase-3 3, 6, and -7, which ultimately leads to apoptosis [10].

Additionally, crosstalk occurs between the extrinsic and intrinsic apoptotic pathways. This occurs through caspase-8, which can interact with Bid. This interaction further promotes the activation of Bax and Bak, which promotes apoptosis in a manner previously described [38,54].

### 2.4. Anti-Proliferative Effects of Curcumin

Curcumin can exert anti-proliferative effects by arresting the cell cycle and promoting apoptosis in multiple cancer cell lines. A recent study found curcumin arrested cell growth and induced apoptosis in an MCF-7 breast cancer cell line by activating p21 and poly(ADP-ribose) polymerase (PARP). Both p21 and PARP are influenced by pathways curcumin is known to interact with, including MAPK, NF-*κ*B, Nrf2, and p53 pathways [55,56]. In a similar study, curcumin caused G2/M cell cycle arrest by affecting the ataxia-telangiectasia mutated (ATM)/Chk1/p53 axis [57]. Curcumin is also implicated in interfering with PI3K/Akt and STAT3 pathways and multiple cyclins, which results in increased cellular arrest and reduced proliferation [58,59,60].

To explore this further, curcumin has been shown to directly inhibit PI3K as well as Akt phosphorylation, which prevents the release of pro-survival signals and promotes apoptosis [47]. Within the MAPK pathway, curcumin has been shown to modulate ERK, c-Jun N-terminal kinase (JNK), and p38 in ways that promote apoptosis. By inhibiting ERK and promoting JNK and p38, apoptosis can be increased [61]. Finally, curcumin has been shown to inhibit STAT3 signaling, which produces cyclin D1 and c-Myc downstream. Both products play crucial roles in cell cycle progression, and by inhibiting them, cell cycle arrest and apoptosis are promoted [62].

Curcumin also targets cancer stem cells by modulating Wnt/β-catenin [63], notch [64], and hedgehog (Hh) pathways [65]. One study by Dou et al. found that curcumin inhibited the Wnt/β-catenin pathway compared to controls by inhibiting miR130-a and expressing negative regulators such as Nkd2 [66]. By targeting cancer stem cell pathways, curcumin may have the potential to slow the proliferation and progression of cancer cells.

Notch receptors are involved in cell differentiation, proliferation, and apoptosis. Cancer can utilize these receptors to promote cancer progression, but curcumin has been shown to decrease notch receptor activity. These receptors have extracellular, transmembrane, and intracellular domains. When a ligand, such as delta-like or Jag, binds to the extracellular domain, there is a conformational change that cleaves the intracellular domain. The intracellular domain then translocates to the nucleus, promoting notch gene expression. Notch translocation increases the transcription of notch genes, including NF-*κ*B, VEGF, and angiogenic genes [67].

In one study, low doses of curcumin were associated with decreased activity in the notch pathway in multiple cell lines [64,68]. The mechanism by which curcumin affects notch pathways is not well studied, and more research is necessary to clarify curcumin’s role. However, one study notes that curcumin may affect the notch intracellular domain’s ability to bind to DNA and affect notch gene expression [69]. Another study demonstrated that curcumin could downregulate the expression of *γ*-secretase, which plays a role in cleaving the intracellular domain [70]. While it seems curcumin plays a role in decreasing the activity in notch-related pathways, further insights are necessary to fully understand this relationship.

Finally, certain cancers can upregulate Hh pathways to proliferate and grow [65,71]. A growing body of literature is beginning to indicate that curcumin can downregulate Hh pathways to slow the progression of certain cancers. In a study of pancreatic cancer, curcumin decreased the expression of Hh compared to controls, which slowed the progression of cancer [71]. In another study, curcumin decreased the expression of Hh-related proteins such as glioma-associated oncogene homolog-1 (GLI1) in medulloblastoma cells, indicating that curcumin downregulated Hh pathways [65]. GLI1 is the final effector protein in Sonic Hh receptors, and it translocates to the nucleus and regulates the expression of genes involved in cell proliferation, differentiation, and survival [72]. Additionally, curcumin may be able to sensitize certain tumors to radiotherapy, especially in gliomas, by inhibiting Hh signaling pathways [73]. Further evidence is necessary to understand curcumin’s full potential in inhibiting Hh pathways. Preliminary evidence shows that curcumin has a positive effect in preventing cancer progression and potentially sensitizing cancer cells to treatment.

### 2.5. Inhibition of Angiogenesis and Metastasis

One hallmark of cancer is its ability to become less dependent on typical mechanisms that regulate growth and development. Cancer promotes angiogenesis to supply itself with blood, which increases its ability to receive nutrients and increases the risk of metastasis [74]. Cancer can promote angiogenesis in multiple ways, including increased expression of VEGF, fibroblast growth factors (FGF), and MMPs. Curcumin has been shown to modulate each of these to prevent angiogenesis in cancer [11].

FGF promotes the transcription of cyclin D1, cMyc, MMPs, and VEGF, among others, and has downstream impacts on Ras/MAPK, PI3K/Akt, and STAT pathways [75,76,77]. Curcumin was shown to inhibit the transcription of FGF in an MDA-MB-231 cell line. This study provides preliminary evidence that curcumin could work as an anti-FGF agent [78]. Additionally, another study demonstrated curcumin could target the FGF-2 angiogenic signaling pathway to inhibit MMP-9, which is responsible for tissue remodeling relating to the growth of new blood vessels [79]. While more research is necessary to establish curcumin’s full effects on FGF, preliminary evidence suggests curcumin inhibits FGF and slows angiogenesis.

COX-2 is an enzyme involved in the production of prostaglandins, which signal to produce pro-angiogenic actions such as vascular sprouting, migration, tube formation, VEGF production, and enhanced endothelial cell survival through Bcl-2 and Akt pathways [80]. Curcumin has been demonstrated to reduce the COX-2 and VEGF angiogenic biomarkers in hepatocellular carcinoma cancer and reduce angiogenesis [81]. In another study of colon cancer, curcumin was shown to inhibit the mRNA and protein expression of COX-2 [82]. A final study found that curcumin’s ability to affect COX-2 enzyme activity is unrelated to its ability to suppress transcription of COX-2, indicating curcumin’s effect on COX-2 is multifaceted [83]. Further evidence is necessary to fully understand the mechanism of curcumin’s effect on COX-2 and angiogenesis. A growing body of literature indicates that curcumin can inhibit COX-2, which results in decreased angiogenesis in multiple cancers.

As previously mentioned, curcumin has been shown to reduce VEGF [81]. VEGF expression is mediated via the PI3K/Akt signaling pathway, which is downregulated by curcumin. In one study, both the protein and mRNA expression of VEGF, PI3K, and Akt were downregulated by curcumin [84]. Additionally, as previously mentioned, curcumin can directly interact with Akt, PI3K, and other growth factors, including EGFR [45,46], but it also indirectly affects this pathway by inhibiting IκKβ, which is an upstream regulator of mTORC1. This results in inhibition of the PI3K/Akt/mTOR pathway [16,47]. Inhibition of the PI3K/Akt pathway promotes apoptosis because Akt can no longer phosphorylate the BAD protein at Ser 136, releasing the pro-survival Bcl-2 molecule [48,49]. By inhibiting VEGF, curcumin is also implicated in interfering with the expression of angiopoietin 1 and 2 gene expression in multiple cancer cell lines, which is another downstream method curcumin may inhibit angiogenesis [85]. Curcumin’s ability to reduce VEGF makes it an interesting compound for multiple medical conditions beyond cancer. While this is beyond the scope of this review, this highlights that the anti-angiogenic effects of curcumin may be more broadly utilized.

Finally, curcumin can inhibit properties of the epithelial–mesenchymal transition (EMT) in cancer. The EMT has been heralded as a key attribute in carcinogenesis because it is highly conserved and enhances mobility, invasion, and resistance to apoptosis [86]. One study examined curcumin’s impact on breast cancer; they found that EMT-related genes, such as E-cadherin, N-cadherin, and fibronectin, among others. These changes led to a decrease in the migratory and invasive capabilities of the cancer cells [87]. Interestingly, curcumin has been studied for its ability to upregulate miRNA and DNA methylation in the EMT, which prevents the transcription of EMT-associated genes. Additionally, curcumin has been shown to downregulate PI3K/Akt/mTOR, NF-*κ*B, and transforming growth factor-β (TGF-β), which are all pathways associated with the proliferation of EMT cells [88]. An overview summary of curcumin’s effects can be seen in Table 1.

## 3. Curcumin and the Immune System

### 3.1. T Cells

T cells are involved in immunosurveillance and can recognize and eliminate abnormal cells within the body. Cancer cells can evade these cells by upregulating T-cell inhibitors such as programmed death-1 ligand-1 (PD-L1), which has led to many immunotherapies focused on reactivating normal T-cell function [89]. Curcumin’s ability to target major immunomodulatory pathways such as NF-*κ*B means it may be useful in restoring T-cell function in cancer or modulating the environment to boost immunotherapy efficacy.

Curcumin has been shown to impact the activation and proliferation of T cells by interfering with activator protein-1 (AP-1) and NF-*κ*B signaling. Naïve T cells differentiate into different types of effector cells based on cytokine profiles and immunomodulatory functions. To start this process, the T-cell receptor must be activated, subsequently activating Nuclear Factor of Activated T cells (NFAT), AP-1, and NF-*κ*B pathways. Later in this process, AP-1 combines with NFAT at IL-2 promoter sites, which enhances DNA binding and transcription of cytokines, notably IL-2 [12]. Curcumin has been demonstrated to suppress AP-1 transcription factors by suppressing the MAPK pathway [90]. Curcumin also inhibits NF-*κ*B, which has been associated with reduced survival of both normal and cancer cells. NF-*κ*B promotes genes important to T-cell differentiation and IL-2 production [16]. This evidence indicates curcumin makes it more difficult to activate and differentiate T cells, which inhibits their ability to attack cancer cells. While this seems contradictory to the notion that curcumin could be used as an adjuvant, there are redeeming qualities.

Curcumin’s effect on NF-*κ*B is context-dependent. Under high oxidative stress, as can occur in the TME, NF-*κ*B signaling can become suppressed beyond desirable levels. Curcumin has been shown to maintain NF-*κ*B signaling homeostasis, despite normally inhibiting this pathway [91]. Interestingly, this seems to make T cells more resistant to apoptosis. This also may improve the infiltration of T cells into the TME, making curcumin an interesting adjuvant in ICI therapy [92]. To summarize, curcumin seems to prevent the activation and differentiation of new T cells. However, for T cells that are already operational, curcumin seems to be beneficial. This property makes curcumin interesting as a potential adjuvant in adoptive T-cell therapy.

Regulatory T cells (Tregs) are a specialized subset of T cells that play a role in suppressing the immune system to prevent an overreactive immune response. They are characterized by the expression of FoxP3, a transcription factor important for their development and function. Tregs also express cytotoxic T-lymphocyte-associated antigen-4 (CTLA-4), which is a molecule that has been targeted by ICI because of its ability to inhibit crucial immune cells [93]. In cancer, Tregs become overproduced, and they then overexpress CTLA-4, which inhibits the activation and proliferation of effector T cells [94]. A growing number of studies have suggested that curcumin can decrease Tregs by converting them to Th1 cells by increasing the expression of interferon-*γ* (IFN-*γ*) [95]. Simply put, IFN-*γ* binds to a receptor on the Treg cell, initiating a JAK/STAT1 cascade, ultimately leading to the transcription of genes associated with Th1 cell differentiation [96].

### 3.2. Macrophages

The TME is a critical tumor growth, progression, and metastasis regulator. Within the TME, immune cells can become modulated to favor cancer progression as seen in tumor-associated macrophages (TAMs). Macrophages can be polarized into either M1 or M2 macrophages. M2 macrophages exhibit pro-cancer effects, including angiogenic, immune-suppressive, and hypoxic activities, whereas M1 macrophages suppress tumor growth. Therefore, it is crucial to develop therapeutics that can promote M1 or inhibit M2 macrophage production [91]. The literature suggests curcumin favorably affects TAMs, as visualized in Figure 3.

Many studies have been published on curcumin’s ability to impact the M1/M2 macrophage paradigm, which plays a role in tumor progression. Cancer cells often promote M2 macrophages by fostering a TME favorable to that phenotype [97]. Curcumin has been shown to reverse this by inhibiting IL-1β, TNF-⍺, C-C chemokine receptor-7 (CCR7), and iNOS, which are associated with the production of M2. This prevents the overproduction of M2 macrophages and maintains homeostasis in the TME [98].

Curcumin has also been shown to inhibit M1 macrophage polarization through the toll-like receptor-4 (TLR-4) mediated signaling pathways inhibition. While this seems counterproductive, M1 macrophage overproduction is associated with chronic inflammatory conditions. The ability to decrease the M1:M2 ratio may be beneficial under certain conditions, such as a highly inflammatory TME. Curcumin can mitigate extreme macrophage polarization and return the TME to homeostasis [99]. This serves as another example of curcumin being a homeostatic agent. This was previously seen in TF-*κ*B signaling pathways in T cells.

Modified curcumin molecules have been shown to more potently target macrophage polarization than normal curcumin. This shows the potential of curcumin analogs to target certain aspects of the TME [13]. This knowledge can be potentially used to modify the TME in a manner that is favorable for immunotherapies.

### 3.3. Dendritic Cells

Dendritic cells (DC) are antigen-presenting cells that capture antigens from their surroundings, process them, and present antigens to lymphocytes to initiate and regulate the immune response [100]. Dendritic cells are critical in initiating and sustaining an effective T-cell-mediated anti-tumor immune response. In this sense, DC prime T cells act. Additionally, the function of DC within the TME may determine the efficacy of immunotherapies [101]. In a cancer context, curcumin effects are context-dependent. Curcumin blocks DC maturation markers, cytokines, and chemokines in DC, which disrupts their antigen-presenting activity. This makes DCs resistant to immunostimulants and reduces the expression of co-stimulatory molecules on DCs. Inhibition of NF- *κ*B, AP-1, and MAPK pathways is implicated in curcumin’s ability to have these effects [14]. Alternatively, within the cancer TME, DCs can become tolerogenic, which is driven by chronic inflammation and tumor-derived signals. Once the DCs become tolerogenic, they cannot respond to the cancer and therefore become pro-tumorigenic. In this context, curcumin can be beneficial and restore the function of DCs by blocking the major pathways necessary for DC suppression, including NF-*κ*B, AP-1, and MAPK [14,101]. This context-dependent effect should be considered in therapy.

### 3.4. Natural Killer Cells

NK cells are a type of lymphocyte distinct from T and B cells. These cells participate in immunosurveillance and cytokine production in the innate immune system [102]. They possess a potent ability to directly kill foreign or damaged cells, making them important for anti-tumor immunity. The TME within cancer can disable NK cells by providing a hypoxic, acidic, or glucose-deprived environment. The TME can also inhibit TGF-β, which disrupts NK cell functionality [103]. Specifically, TGF-β has been found to decrease the expression of activating receptors (NKG2D and NKp30) on NK cells, which impairs their cytotoxic ability and anti-tumor activity [104].

A growing body of literature shows that curcumin can enhance the activity of NK cells. One study of breast cancer showed that curcumin increased the expression of CD56 and CD16, which are key surface proteins in NK cells. These proteins are closely associated with NK function and activity [15]. Additionally, under certain circumstances, curcumin increases nitric oxide production by stimulating iNOS, which contributes to the tumoricidal effect of NK cells. This occurs when NF-*κ*B signaling is suppressed beyond desirable levels [105]. Finally, curcumin can increase the cytotoxicity of NK cells, leading to an increase in apoptosis [106].

Curcumin also promotes IFN-*γ*, which subsequently increases key CD16 and CD56 proteins in NK cells [15]. IFN-*γ* is a critical cytokine that plays a role in enhancing the function of NK cells. Once IFN-*γ* binds to its receptor, STAT1 recruitment occurs, which results in nuclear translocation and DNA binding. This induces the transition of cytokines, chemokines, and other molecules that can enhance the cytotoxic capabilities of NK cells. These proteins include perforin, granzymes, and CD16 [15,107]. Overall, curcumin positively impacts NK cells and their ability to attack cancer cells. Additionally, a summary of curcumin’s effects on immune cells can be seen in Table 2.

## 4. Curcumin in Combination with Immunotherapy

### 4.1. Curcumin and ICIs

Curcumin targets pathways in cancer that may have the potential to synergistically work with ICIs, which is a type of immunotherapy. Cancer can escape immune detection and attack by upregulating inhibitory immune signals. ICIs can stop the inhibitory signals and allow the immune system to respond normally to cancer cells, leading to their destruction. Current ICIs can target programmed death ligand-1 (PD-L1), a ligand for the PD-1 receptor, which ultimately decreases CD8 T-cell-mediated tumor response in cancer. Other ICIs can inhibit CTLA-4, which is an inhibitory receptor on T cells that downregulates T-cell activity and prevents T-cell activation [108]. Other ICI targets are currently being researched; however, the two previously mentioned are the two most common clinical ICIs.

There is a growing amount of literature that indicates curcumin may be useful in combination with PD-L1 ICIs such as ipilimumab. One study by Lim et al. found that curcumin can destabilize PD-L1 in cancer cells by inhibiting NF-*κ*B and its downstream effects, specifically the CSN5 transcription factor. CSN5 normally inhibits the ubiquitination of PD-L1; therefore, curcumin restores the proper ubiquitination of PD-L1. This also sensitizes cancer cells to anti-PD-L1 and CTLA-4 immunotherapies [109]. Downstream effects of NF-*κ*B are also inhibited, including IL-10, which is associated with decreased cytotoxicity of CD8+ T cells. Curcumin has also been demonstrated to decrease the expression of TGF-β, which is also associated with the decreased cytotoxicity of CD8+ T cells; therefore, their suppression may boost the therapeutic effects of PD-L1 ICIs [110,111]. Finally, curcumin’s ability to downregulate TGF-β was shown to improve the efficacy of PD-L1 inhibitors in a hepatocellular carcinoma cell line [112]. These studies set the groundwork for examining curcumin’s effect in addition to ICIs in clinical trials.

STAT3 pathways in cancer have been associated with immune-resistant mechanisms. Previous studies have demonstrated that STAT3 is inhibited by curcumin [7,9,113]. By directly inhibiting STAT3, the production of IL-6 and IL-8 has decreased in multiple cancer cell lines [113]. Therefore, curcumin may be able to decrease the production of IL-6 and IL-8, which are associated with chemoresistance, tumor growth, and metastasis [114]. Additionally, despite typically harming DC, curcumin was surprisingly shown to restore DC function by targeting STAT3 and reducing IL-6 production in certain contexts. This was also shown to enhance T-cell induction and improve the efficacy of PD-L1 ICIs [113]. This demonstrates curcumin’s potentially synergistic effect with ICIs by targeting STAT3 pathways.

Curcumin has also been shown to reduce the expression of CTLA-4 at both protein and mRNA levels. By decreasing the expression of CTLA-4, inhibitor signals cannot be sent to T cells, which prevents the tumor from escaping the immune response [115]. Decreased levels of CTLA-4 have also been associated with increased efficacy of anti-CTLA-4 ICIs [116].

Finally, curcumin derivatives have been demonstrated to act similarly. One study demonstrated that bisdemethoxycurcumin in combination with PD-L1 blockades significantly increased CD8+ T-cell tumor infiltration, increased IFN-*γ*, and granzyme B. These changes were associated with increased survival in nude mice that had bladder cancer metastasis [117]. This is potentially an interesting area of research. By developing new curcumin derivatives, there may be a potential increase in these synergistic effects with ICIs or even customizing the response that is needed.

There is currently a lack of evidence involving the use of curcumin in conjunction with immunotherapies in any clinical setting. This represents a major knowledge gap to date, and future studies should consider these combinations.

### 4.2. Curcumin and Adoptive Cell Therapy

Adoptive cell therapy (ACT) involves extracting, modifying, and reinfusing immune cells to treat cancer. This method allows research to boost one’s immune system to recognize and attack cancer. ACT can be deployed in a couple of ways, including tumor-infiltrating lymphocyte therapy, engineered T-cell receptor therapy, chimeric antigen receptor T-cell (CAR T) therapy, and NK cell therapy. In tumor-infiltrating lymphocyte therapy, T cells are taken from the patient’s tumor and multiplied, then activated. They are then reinfused, where they can efficiently destroy tumors. In engineered T-cell receptor therapy, the same process occurs, except a new T-cell receptor is added, enabling them to target specific cancer antigens. CAR T cell therapy equips a patient’s T cells with a synthetic receptor (chimeric antigen receptor). This allows T cells to bind to cancer cells even when antigens are not presented to them. NK cell therapy utilizes the same techniques but with NK cells [118]. This method allows researchers to modify and customize the T-cell response to cancer cells.

We propose three mechanisms by which curcumin may improve adoptive cell therapies (ACT), including CAR T therapy: Curcumin may reduce the immunosuppressive burden, protect CAR T cells from apoptosis and exhaustion, and it may enhance their intrinsic cytotoxic capabilities. One challenge with ACT is that the effects are often transient and prone to failure due to the immunosuppression within the TME. Curcumin may increase the efficacy of ACT by increasing CD8+ T-cell activation by inhibiting TGF-β, a major immunosuppressive cytokine in the TME [119]. Further research should focus on curcumin’s ability to increase CD8+ T-cell resistance to apoptosis and whether that correlates with increased ACT efficacy. Additionally, given curcumin’s multifaceted manner of attacking the hallmarks of cancer and improving TME conditions for T cells, more research should be conducted to explore whether curcumin can modify the TME in advantageous ways for ACT.

Additionally, curcumin may protect ACT cells from apoptosis through its antioxidant effects previously laid out. An immune cell becomes exhausted when it is chronically exposed to inflammatory cytokines, including IL-2, IFN- *γ*, and TNF-⍺, which can be prevalent in the TME. Once this cell is exhausted, it will upregulate the expression of inhibitory receptors, thereby making it more likely that the cell will become inactivated or undergo apoptosis [120]. Curcumin may improve the efficacy of ACT by reducing the production of these pro-inflammatory cytokines in vitro by interacting with NF-*κ*B, STAT1, and STAT3 pathways, which have previously been covered in this paper [121].

Finally, curcumin may increase CAR T cells’ cytotoxicity. CAR T cells kill targeted cancer cells primarily by releasing cytotoxic granules, including perforin and granzymes. These molecules have the power to induce apoptosis in targeted cells [122]. Curcumin has been shown to upregulate the expression of granzyme B activity, thereby potentially increasing the cytotoxicity of CAR T cells [117]. Additionally, curcumin has been shown to increase cytotoxicity of CAR T-cells in a Nalm-6 leukemia model in a dose-dependent manner [121]. Further evidence should be conducted to verify the relationship between CAR T cell efficacy and curcumin.

### 4.3. Curcumin and Cancer Vaccinations

Cancer vaccinations come in two varieties: preventative and therapeutic. Preventative vaccines allow the immune system to recognize an antigen and activate the immune system before deleterious effects can take hold. Current examples of preventative vaccinations include the HPV vaccinations (Gardasil and Cervarix) and the hepatitis B vaccination [123]. Therapeutic vaccinations can boost the immune system’s attack on cancer cells. Response rates on these vaccines can vary because they rely on specific peptide antigen complexes, which can vary from patient to patient [124].

Curcumin may act as an adjuvant for therapeutic cancer vaccines by remodeling the TME. In one study, a curcumin polyethylene glycol conjugate (CUR-PEG) was delivered intravenously to a melanoma tumor, followed by a Trp2 peptide vaccination. The researchers found that CUR-PEG and Trp2 vaccines resulted in a synergistic anti-tumor effect by boosting T-lymphocyte response by 41% and IFN-*γ* by sevenfold. Additionally, researchers found that Treg and MDSCs were inhibited along with other pro-inflammatory cytokines [125]. Curcumin interacts with NF-*κ*B, STAT1, and STAT3 pathways, which reduces the production of pro-inflammatory cytokines [121]. This is one mechanism by which curcumin may exert these synergistic effects.

Listeria-MAGE-B is a vaccination that is primarily used to treat breast cancer by inhibiting IL-6. This vaccine works by targeting IL-6, which is particularly high in breast cancer and is associated with immunosuppressive effects in the TME. Curcumin was shown to be a good adjuvant for this vaccine by targeting by boosting T-cell responses and further inhibiting IL-6 production. The mechanism by which curcumin reduces the production of IL-6 is through its interference with STAT3 pathways [121]. The response of CD4 and CD8 T cells was shown to improve when curcumin was used alongside the vaccination [126].

Finally, curcumin has been shown to boost the efficacy of the FAP⍺c vaccine, which targets fibroblast activation. This vaccination can inhibit fibroblasts and activate the secretion of IFN-*γ*; however, it also induces the expression of indolamine-2,3-dioxygenase (IDO) and TNF-⍺, which contributes to immunosuppression. Curcumin in combination with this vaccination was shown to inhibit IDO and TNF-⍺, which offset the deleterious effects of the vaccination. The authors of this study proposed that this may be an effective vaccination for melanoma treatment [127].

Given curcumin’s ability to attack many aspects of cancer and modulate the immune cells within the TME, it may be useful as an adjuvant in cancer vaccine therapies. As cancer vaccines continue to evolve and progress, further studies should explore curcumin in combination with the therapy.

## 5. Contextual Dualities

In Section 2, there were multiple contextual dualities that were discussed, including NF-*κ*B modulation, T-cell modulation, DCs, and macrophage polarization. In this section, we will further discuss curcumin’s role in these dualities. Curcumin has been shown to be a homeostatic compound that can either increase or decrease the activation of NF-*κ*B. When the NF-*κ*B pathway becomes too active, curcumin will inhibit it by mechanisms previously discussed. This inhibition unlocks the pro-apoptotic effects of curcumin in various cancers [16]. When there is high oxidative stress, NF-*κ*B signaling becomes suppressed. Curcumin in this context will promote NF-*κ*B signaling by preventing its suppression. This is likely due to curcumin’s powerful antioxidant ability [91,92]. This effect highlights curcumin’s ability to maintain proper levels of NF-*κ*B signaling in different adverse environments in a cancer context.

Curcumin also leads to a core dichotomy in T-cell modulation. Curcumin can suppress the activation and differentiation of naïve T cells by interfering with AP-1 and NF-*κ*B signaling pathways. These pathways are key for T-cell activation and IL-2 production, and their inhibition is detrimental to immune initiation [12,16,90]. Conversely, studies have indicated curcumin can enhance the survival and infiltration of T cells in the TME, which boosts the immune response against cancer [91,92]. Additionally, curcumin has been shown to convert immunosuppressive Tregs to a Th1 phenotype, which increases IFN-γ. This then promotes a more robust immune response against cancer cells [95]. Finally, curcumin has been demonstrated to destabilize PD-L1, reduce CTLA-4 expression, and modulate STAT3 and TGF-β. Each of these molecules is immunosuppressive against T cells. Therefore, their inhibition promotes a T-cell response against cancer cells [109,110,111,113,115].

The dichotomy presented between curcumin and T-cell function can be navigated in the context of immunotherapies. While curcumin suppresses naïve T-cell activation and differentiation, immunotherapies including ACT and ICIs rely on already operational T cells. In these therapies, ACT and ICIs are not concerned with naïve T cells. Conversely, curcumin still will enhance T-cell survival, infiltration, and anti-tumor remodeling of the TME [91,92,109,110,111,113,115,119,121]. With respect to cancer vaccines, further studies should be performed to test whether the TME modulation outweighs the T-cell activation dampening. In these studies, dosing and bioavailability should be considered to further optimize this question.

Within the TME, DCs are critical for a proper immune response. Curcumin has been shown to impede DCs by inhibiting key maturation and activation pathways, including NF-*κ*B, AP-1, and MAPK [14,101]. Alternatively, cancers can commonly co-opt DCs and cause them to become tolerogenic, which prevents a proper immune response in the TME. In these contexts, curcumin may prevent this immunosuppression [101]. Further studies are necessary to elucidate the role of curcumin and DCs in a cancer context.

Curcumin also impacts macrophage polarization. It has previously been discussed that macrophages can adopt an M1 or M2 phenotype, with M1 being an anti-cancer phenotype and M2 being a pro-tumorigenic phenotype. Curcumin has been shown to inhibit the M2 phenotype while having a homeostatic effect on the M1 phenotype [98,99]. This is the best outcome because when the M1 phenotype becomes overabundant, curcumin can inhibition their formation, allowing for an optimal M1:M2 ratio [99].

## 6. Challenges and Future Perspectives

The major problem with using curcumin as an adjuvant in immunotherapies is its poor pharmacokinetic profile. Curcumin is rapidly metabolized and absorbed poorly, which means dietary consumption of curcumin will not maximize its therapeutic potential. Curcumin is chemically stable between a pH of 1 and 6, but it is insoluble in water at this pH range. At physiological pH, it undergoes an autoxidation reaction, which leads to a series of dicyclopentadiene products. Additionally, curcumin binds to enterocyte proteins that can modify its structure. These properties highlight curcumin’s less-than-ideal pharmacokinetic profile [128]. This demonstrates that under physiological conditions, only a small percentage of curcumin can exert its desirable effects.

There have been recent efforts to find delivery methods that preserve the chemical structure of curcumin in a biologically available form. One of the main strategies for increasing curcumin’s bioavailability is to utilize agents that inhibit or delay metabolism or to find formulations that offer longer circulation, improved permeability, or more resistance to metabolic processes [129]. There are three generations of methods that have been used to increase bioavailability in the manner previously discussed [128].

First-generation formulations use natural agents in addition to curcumin [128]. Piperine has been shown to potently increase the absorption of curcumin. Piperine is a natural alkaloid of black pepper, and one study demonstrated that 2 g of curcumin plus 5 mg of piperine displayed a 3-fold increase in curcumin absorption [130]. Other studies have also shown that curcumin, in addition to piperine, increases the absorption of curcumin [131]. Curcumin–phospholipid complexes have shown similar results to piperine [132].

Second-generation formulations demonstrated a large improvement over first-generation agents by mixing curcumin with hydrophilic nanoparticles. One study on Theracumin, which is a water-soluble formulation that contains polyvinyl pyrrolidone, cellulosic derivatives, and other antioxidants, has shown a 40-fold increase in the concentration of curcumin [133]. Other strategies to improve curcumin bioavailability include lipid-based formulations and micellar systems [129].

Today, there are novel third-generation formulations, which mix curcumin non-covalently with food-grade excipients. This is thought to maximize absorption. One example of this is Curcuwin Ultra+, which has been shown to increase curcumin bioavailability by 64.7 times higher than oral curcumin [134]. These findings show marked advancements; however, the delivery systems of curcumin remain of interest. Despite these advancements, these developments have not been widely used in the literature.

While there are no published clinical trials exploring the link between curcumin and immunotherapies to our knowledge, there are clinical trials involving other interesting uses for curcumin in the treatment of various cancers. We have provided a summary of ongoing and completed clinical trials involving the various uses of curcumin in different cancer contexts seen in Table 3. Generally, these trials indicate curcumin is well tolerated in most contexts with limited adverse events reported. Additionally, Table 3 provides the available clinical evidence of the new generation curcumin formulation.

Finally, certain areas of the literature currently require more research. First, curcumin’s effects on adoptive cell therapies should be studied more extensively because there is a lack of studies on the matter. Given curcumin’s ability to remodel the TME and its preliminary success with other immunotherapies sets the groundwork for a study including ACT. Additionally, more clinical trials on curcumin should be conducted. Many studies on this phytochemical rely on cell lines and mouse models. However, due to the low risk and established benefits in these models, more clinical trials may be warranted. Finally, it would be interesting to study curcumin analogs, which may be able to hone certain properties as seen in bisdemethoxycurcumin’s effect on infiltrating T cells [117].

## 7. Conclusions

Curcumin has a well-established body of evidence showing its anti-cancer effects. These effects seem to include anti-apoptotic, oxidant, inflammatory, proliferative, and angiogenic properties. The ability to impact the NF-*κ*B pathway seems to be implicated in many of these effects. In addition, curcumin also affects major pathways implicated in cancer, including PI3K/Akt, Bcl2, JAK/STAT, and p53 pathways. By modulating these pathways, curcumin can modulate the TME and immune cells to promote an anti-cancer response. Curcumin seems to benefit T cells, macrophages, NK cells, and DC cells. Additionally, some of curcumin’s effects seem to be context-dependent. For example, this was seen in NF-*κ*B, we highlight these dichotomies and how they can be navigated in the paper. These effects should be considered when using curcumin therapeutically. These properties have also been demonstrated to improve the efficacy of certain immunotherapies such as ICIs. More literature is necessary to conclude that curcumin increases the efficacy of ACT and cancer vaccinations; however, early indications are promising. Further research on improving bioavailability and curcumin analogs could improve the therapeutic benefit.

## Figures and Tables

**Figure 1 cancers-17-02491-f001:**
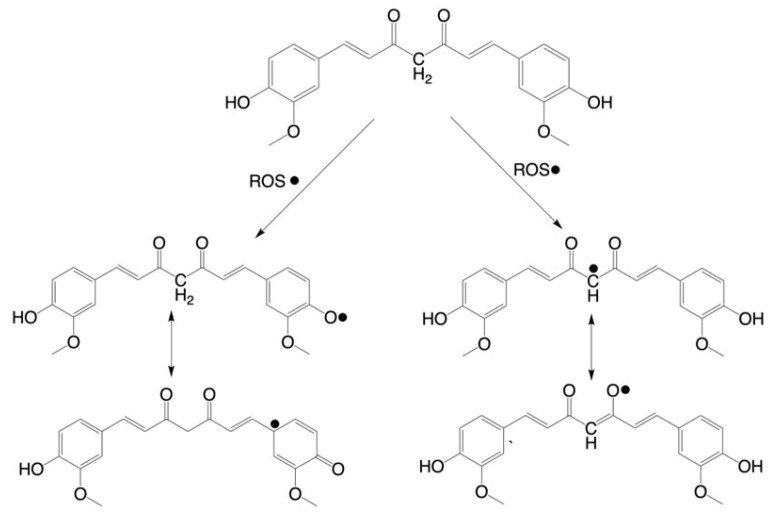
Curcumin neutralizes ROS by two proposed mechanisms seen in the figure. The first mechanism is through donating a hydrogen atom to ROS. The second mechanism is by donating hydrogen from the carbon on the β-diketone group to neutralize ROS. The radical that is created in both pathways can be neutralized through further reactions.

**Figure 2 cancers-17-02491-f002:**
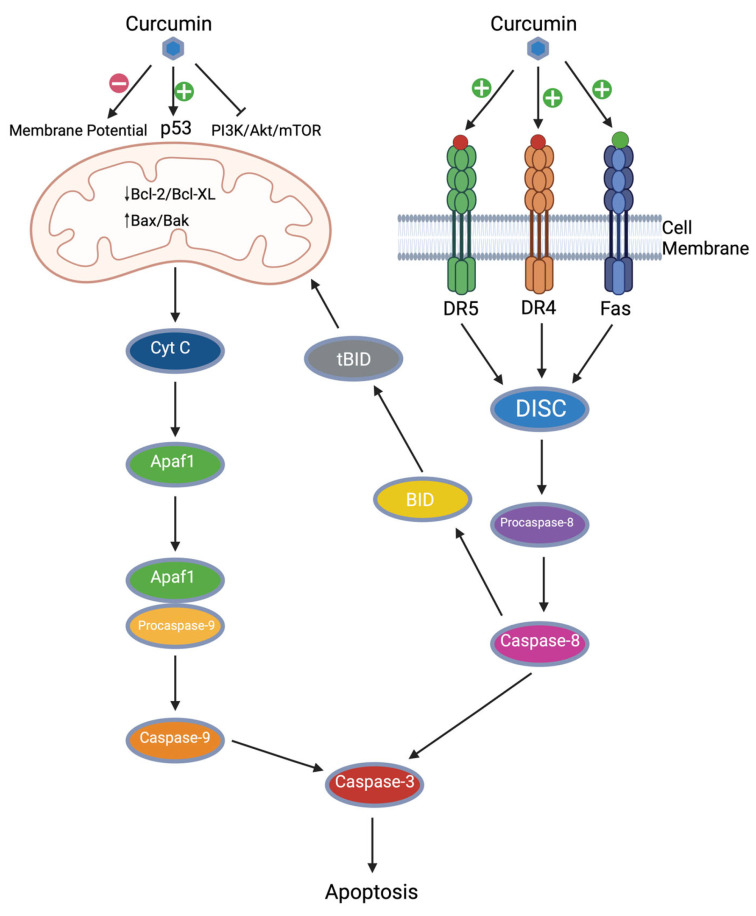
Curcumin promotes intrinsic apoptosis by directly diminishing the mitochondrial membrane potential, promoting the p53 tumor suppressor, and inhibiting the PI3K/Akt/mTOR growth pathway. Curcumin also promotes extrinsic apoptosis by increasing the expression of death receptors DR4, DR5, and Fas.

**Figure 3 cancers-17-02491-f003:**
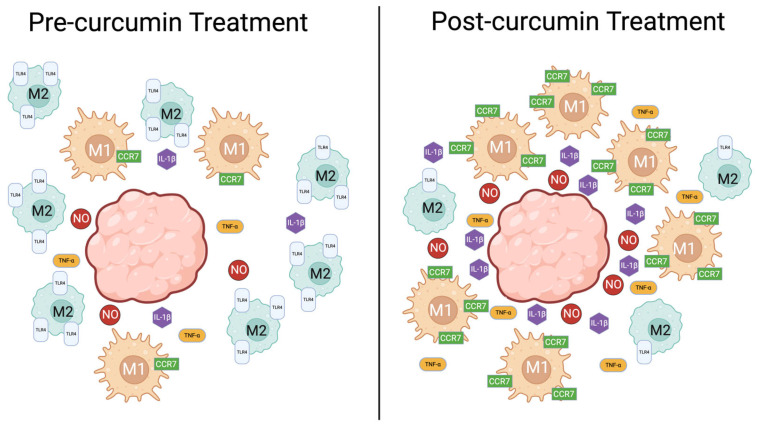
Curcumin affects macrophage polarization within the TME by increasing the M1:M2 ratio. This results in an anti-cancer TME with increased NO, IL-1β, TNF-α, and other cytokines.

**Table 1 cancers-17-02491-t001:** Summary of the downstream effects of NF-*κ*B inhibition by curcumin.

Downstream Effect	Downstream Effect	Result	Reference
Anti-Inflammatory	↓ TNF-⍺	↓ Inflammation	[8,19]
	↓IL-1	↓ Inflammation	[8,19]
	↓ IL-6	↓ Inflammation	[8,19]
	↓ IL-8	↓ Inflammation	[8]
	↓ COX-2	↓ Inflammation	[8,19]
	↓ iNOS	↓ Inflammation	[20]
Apoptosis	↑ Bax	↑ Apoptosis	[31,36,38,39,40]
	↑ Bak	↑ Apoptosis	[31]
	↑ PUMA	↑ Apoptosis	[31]
	↓ Bcl-2	↑ Apoptosis	[31,39,40]
	↓ Bcl-XL	↑ Apoptosis	[31,39,40]
	↑ DR4	↑ Apoptosis	[50,52]
	↑ DR5	↑ Apoptosis	[50,52]
Proliferative	↑ G2/M Arrest (via ATM/Chk1/p53)	↓ Cancer Cell Proliferation	[7,47,60]
	↓ PI3K/Akt/STAT3	↓ Cancer Cell Proliferation	[47,59,78]
Angiogenic	↓ VEGF	↓ Angiogenesis	[85]
	↓ FGF	↓ Angiogenesis	[78]
	↓ MMP-9	↓ Angiogenesis	[78]
	↓ COX-2	↓ Angiogenesis	[84]
	↓ PI3K/Akt/mTOR	↓ Angiogenesis	[78]

**Table 2 cancers-17-02491-t002:** Summary of curcumin’s effects on immune cells.

Immune Cell Type	Subtype	Curcumin’s Effect	Mechanism	Reference
T Cells	Naïve T Cells	Inhibits activation and differentiation	Suppresses AP-1 and NF-*κ*B signaling. Reduces IL-2 production and T-cell differentiation	[12]
	Effector T Cells	Enhances resistance to apoptosis	Maintains NF-*κ*B signaling homeostasis under oxidative stress	[12,91,92,94]
	Regulatory T cells	Converts Tregs to Th1 cells	Increases expression of IFN-*γ*, activating Jak/STAT1 cascade	[93,94,95,96]
Macrophages	M1 Macrophages	Inhibits M1 polarization	Inhibits TLR-4-mediated signaling pathways	[99]
	M2 macrophages	Inhibits M2 production	Inhibits IL-1β, TNF-⍺, CCR7, and iNOS preventing overproduction of M2 macrophages	[97,98]
Dendritic Cells	Immature DCs	Inhibits maturation and antigen presentation	Blocks maturation markers, cytokines, and chemokines, including NF-*κ*B, AP-1, and MAPK pathways	[14,100,101]
	Mature DCs	Reduces immunostimulatory activity	Decreases expression of co-stimulatory molecules	[14,100,101]
Natural Killer Cells		Enhances activity and cytotoxicity	Increases expression of CD56 and CD16, stimulates iNOS and nitric oxide production, and promotes IFN-*γ*	[15,104,105,106,107]

**Table 3 cancers-17-02491-t003:** Clinical trials involving curcumin and immunotherapies.

Trial Type	Trial Status	Cancer Focus	Intervention	Key Takeaways	Reference
NCT06626230 (Phase 1)	Ongoing	Safety	Curcumin via anal suppository	Objective: Trial testing for the best dose of curcumin in HIV patients with anal lesions.	[135]
NCT06080841 (Phase 1)	Ongoing	Cervical Cancer	Oral curcumin	Objective: Does curcumin supplementation increase the levels of p53 and apoptosis in cervical cancer patients, and what is the optimal dose of supplementation.	[136]
NCT06063486 (Phase 2)	Ongoing	Hematologic Cancers	Oral curcumin	Objective: Used to compare the change in inflammatory cytokine levels in patients treated with curcumin vs. placebo, as well as to measure symptomology in these groups.	[137]
NCT06044142 (Phase 1)	Ongoing	Pediatric Cancer	Curcumin vs. photo-biomodulation therapy	Objective: Trial will assess the impact of non-invasive photodynamic therapy with curcumin and photo-bio-modulation low-level laser treatment in managing mucositis induced by chemotherapy in pediatric patients.	[138]
NCT05982197	Completed	Head and Neck Cancers	Topical curcumin against radiotherapy-induced oral mucositis	Outcome: Both curcumin mouthwash and nanocapsules were effective, safe, and well tolerated in treating radiation-induced oral mucositis, allowing for radiation to be better tolerated.	[139]
NCT05966441 (Phase 2)	Ongoing	Breast Cancer	Curcumin in combination with paclitaxel to prevent peripheral neuropathy	Objective: Trial testing for the neuroprotective efficacy of 2 g oral curcumin in breast cancer patients. Peripheral neuropathy is a major side effect of paclitaxel chemotherapy.	[140]
NCT05947513	Ongoing	Cervical Cancer	Curcumin with palliative radiotherapy	Objective: Is adding curcumin to standard-of-care palliative radiotherapy of cervical cancer patients feasible and improve therapeutic responses while maintaining patient safety? To test, 250 mg curcumin capsules will be administered 4 times daily.	[141]
NCT05768919 (Phase I/II)	Ongoing	Gliomas	Liposomal curcumin with radiotherapy and temozolomide	Objectives: This study implements liposomal curcumin with standard radiation and adjuvant temozolomide in high-grade glioma patients to test dosage, safety, and feasibility.	[142]
NCT05688488 (Phase I/II)	Ongoing	Bilateral Vocal Cords	Curcumin use to prevent vocal cord adhesion in laryngeal papilloma surgery	Objectives: This study tests topical application of curcumin on bilateral vocal cords following vocal cord endoscopy to relieve vocal cord adhesion.	[143]
NCT04208334 (Phase IIa)	Completed	Head and Neck Cancer	Curcumin supplementation in Cancer anorexia–cachexia syndrome	Outcome: There was a significant benefit from curcumin in treating cancer anorexia–cachexia syndrome in patients with head and neck cancers by restoring metabolic processes.	[144]
NCT03980509 (Phase 1)	Ongoing	Breast Cancer	Curcumin window trial before surgery	Objective: Can curcumin reduce cancer load for stage I-III invasive breast cancer patients prior to undergoing surgery?	[145]
NCT03211104	Completed	Prostate Cancer	Oral curcumin with intermittent androgen deprivation therapy	Outcome: Patients in the curcumin treatment group had significantly lower PSA progression compared to the placebo group. Additionally, there were significantly less adverse events in the curcumin group. However, curcumin did not reduce the amount of time patients could be off intermittent androgen deprivation.	[146]
NCT03192059 (Phase II)	Completed	Cervical Cancer	Immune modulatory 5-drug cocktail with pembrolizumab and radiation	Outcome: A 5-drug cocktail including cyclophosphamide, aspirin, lansoprazole, vitamin D, and curcumin started 2 weeks prior to radioimmunotherapy, resulted in significantly higher proportion of peripheral T cells in responders compared to nonresponders. Additionally, health-related quality of life was stable over time with acceptable toxicity. This trial failed to meet its primary objective, which was to measure objective response rate at week 26.	[147]
NCT03072992 (Phase II)	Completed	Breast Cancer	Curcumin with chemotherapy	Outcome: Curcumin in combination with docetaxel showed no difference in objective response rate, with a slight tendency toward longer progression-free survival at 12 months, which did not reach significance.	[148]
NCT02782949 (Phase II)	Ongoing	Gastric Cancer	Curcumin	Objective: Test whether patients treated with 500mg of curcumin for 180 days will have a protective effect against gastric cancer.	[149]
NCT02556632 (Phase II)	Completed	Breast Cancer	Topical curcumin in reducing radiation-induced dermatitis	Outcome: This study found that 2% topical curcumin significantly reduced redness and irritation of radiation-induced dermatitis in patients with non-inflammatory breast cancer, indicating curcumin made radiation therapy more tolerable.	[150]
NCT02321293	Completed	Non-small cell lung cancer	Curcumin with tyrosine kinase inhibitors	Outcome: This study’s main goal was to examine feasibility and safety of curcumin treatment.	[151]
NCT02138955 (Phase 1)	Completed	General Cancer	Tolerance of IV liposomal curcumin (Lipocurc)	Outcome: The purpose of the study was to test dosage of IV curcumin. This trial found that 300 mg/m^2 liposomal curcumin was the maximal dose tolerated over 6h. The participants in the study included patients with metastatic tumors.	[152]
NCT02100423 (Phase II)	Completed	Chronic Lymphocytic Leukemia	Curcumin and cholecalciferol	Outcome: This study found that curcumin in combination with cholecalciferol was well tolerated and did not result in disease progression. Additionally, patients receiving daily oral treatment in cycles resulted in response-based continuation possible for up to 2 years.	[153]
NCT02017353 (Phase II)	Completed	Endometrial Cancer	Curcumin	Outcome: This study tested whether curcumin changed biomarkers in endometrial cancer; they found no difference in inflammatory biomarkers with a non-significant trend to improve quality of life. Finally, there was significant interpatient variability in biomarker levels.	[154]
NCT01917890 (Phase I)	Completed	Prostate Cancer	Curcumin and urosolic acid	Outcome: Unknown.	[155]
NCT01740323 (Phase II)	Completed	Breast Cancer	Curcumin (Meriva)	Outcome: This trial treated breast cancer patients undergoing radiation therapy. They found that 500 mg Meriva did not statistically affect markers of inflammation, but they found it may be beneficial for fatigue in women treated with neoadjuvant chemotherapy.	[156]
NCT01333917 (Phase 1)	Completed	Colorectal	Curcumin	Outcome: Participants received 4 g curcumin daily, followed by rectal biopsies to assess curcumin as a chemopreventative agent. No results have been published despite the clinical trial being marked as complete.	[157]
NCT01160302 (Phase I)	Completed	Head and Neck Cancer	Curcumin	Objective: Tested curcumin’s biomarker response in head and neck cancers. Despite data collection being complete, no data has been made available.	[158]
NCT01042938 (Phase II)	Completed	Breast Cancer	Curcumin	Outcome: This study tested whether curcumin could reduce radiation-induced dermatitis in breast cancer patients. They found that 2 g of oral curcumin three times a day resulted in reduced radiation dermatitis severity.	[159]
NCT00927485	Completed	Familial Adenomatous Polyposis	Curcumin	Outcome: 4 g of oral curcumin for 30 days resulted in a significantly reduced number of aberrant crypt foci.	[160]
NCT00295035 (Phase III)	Ongoing	Colon Cancer	Curcumin with gemcitabine and celecoxib	Objective: The primary goal is to assess whether the combination therapy can increase the median time to tumor progression from 2.7 months to 4.0 months in metastatic colon cancer patients.	[161]

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
