# Peer review of "From Spice to Survival: The Emerging Role of Curcumin in Cancer Immunotherapy"

_cancers, 2025, doi:10.3390/cancers17152491_

Round 1

Reviewer 1 Report

Comments and Suggestions for Authors

The review “From Spice to Survival: The Emerging Role of Curcumin in Cancer Immunotherapy” provides a timely and well-organized summary of curcumin’s immunomodulatory effects and its potential as an adjuvant to cancer immunotherapy. It effectively covers key mechanisms and therapeutic synergies. While the manuscript is well-referenced and informative, certain sections would benefit from greater clarity and stronger emphasis on clinical relevance.

This study is insightful; however, it has limitations, including a lack of clinical evidence, limited discussion of conflicting data, unclear immune context-dependence, and insufficient attention to bioavailability and safety, which should be addressed to improve clarity and relevance.

Below are several recommendations to enhance your manuscript:

  1. This study is minimal clinical evidence, unclear immune context-dependence, and insufficient discussion of safety, bioavailability, and conflicting data. Only a few clinical examples (e.g., CUR-PEG [123], FAPα vaccine [125]) are mentioned. Adding a table of ongoing trials combining curcumin with immunotherapies would enhance translational relevance.
  2. This study is insightful but would benefit from addressing immune context, safety, and conflicting data. A table of ongoing curcumin–immunotherapy trials could improve clinical relevance.
  3. The manuscript notes curcumin’s context-dependent effects (e.g., NF-κB, DCs). Adding a brief subsection (e.g., “Contextual Dualities”) to clarify these paradoxes and suggest ways to harness benefits while limiting drawbacks would improve clarity.
  4. The abstract should mention specific immunotherapies (e.g., ICIs, vaccines) and note bioavailability challenges. Reframe DC suppression (Section 3.3) as a potential anti-tolerance mechanism. In Section 4.2, propose how curcumin may enhance CAR-T persistence via TME modulation. Define abbreviations at first use and replace “pleiotropic” with “multimodal” or “multifaceted.”

This review offers a valuable synthesis of curcumin’s role in cancer immunotherapy, with strengths including detailed mechanistic insights and well-organized discussion of combination strategies. However, given the outlined limitations, further clarification and validation are necessary. I recommend revisions before the manuscript is suitable for publication.

Author Response

Comment #1

This study is minimal clinical evidence, unclear immune context-dependence, and insufficient discussion of safety, bioavailability, and conflicting data. Only a few clinical examples (e.g., CUR-PEG [123], FAPα vaccine [125]) are mentioned. Adding a table of ongoing trials combining curcumin with immunotherapies would enhance translational relevance.

Response #1

We agreed that adding a table would improve clinical significance and received this comment from other reviewers. To address this, we created table 3. We included recently completed and active trials involving curcumin and cancer which added close to 30 new clinical citations. There is a lack of clinical trials involving curcumin and immunotherapies however there are many trials exploring curcumin with other therapies in multiple contexts that are included in this table. Additionally, we added a paragraph to section 6. Challenges and Future Perspectives which clarifies what is included in this table.

Comment #2

This study is insightful but would benefit from addressing immune context, safety, and conflicting data. A table of ongoing curcumin–immunotherapy trials could improve clinical relevance.

Response#2

We agreed that we needed to improve the evidence of safety and conflicting data. To address this, we included table 3 which includes recent and ongoing clinical trials involving curcumin in various cancers. Again, there is a lack of clinical trials involving curcumin and immunotherapies but there are trials involving curcumin and other treatment modalities. To further address this comment, we included trials examining the safety of curcumin in different forms and commented on how it was tolerated when it was noted in the study results.

Comment #3

The manuscript notes curcumin’s context-dependent effects (e.g., NF-κB, DCs). Adding a brief subsection (e.g., “Contextual Dualities”) to clarify these paradoxes and suggest ways to harness benefits while limiting drawbacks would improve clarity.

Response #3

We agreed that we needed to address some of the contextual dualities better. To address this, we added a new section (Section 5. Contextual Dualities). This was added as a new section due to other reviewers’ comments as well. In this section we summarize major dualities presented in the paper and we highlight the role of T-cells in curcumin use. Additionally, we highlight how immunotherapies presented in the paper might be impacted by this duality.

Comment #4

The abstract should mention specific immunotherapies (e.g., ICIs, vaccines) and note bioavailability challenges. Reframe DC suppression (Section 3.3) as a potential anti-tolerance mechanism. In Section 4.2, propose how curcumin may enhance CAR-T persistence via TME modulation. Define abbreviations at first use and replace “pleiotropic” with “multimodal” or “multifaceted.”

Response #4

We agreed with everything noted in this comment. We modified the abstract to mention specific immunotherapies and noted bioavailability challenges. Additionally, we thought the idea of reframing DC suppression as a potential anti-tolerance mechanism was a great idea we applauded the creativity. We modified the language and reviewed the citations in section 3.3 to reframe this section. We also did a lot of work to address the comment about section 4.2. We included new citations and presented how curcumin could enhance CAR-T persistence in a more organized and complete way. Finally, we changed pleiotropic to multimodal and tried to ensure our abbreviations were defined at first use.

Reviewer 2 Report

Comments and Suggestions for Authors

From Spice to Survival: The Emerging Role of Curcumin in Cancer Immunotherapy

Overview Curcumin, a natural compound derived from turmeric, exhibits anti-cancer properties by modulating key pathways, enhancing immune responses and synergizing with immunotherapies such as ICIs. Despite its potential, challenges like poor bioavailability require advanced formulations to maximize its therapeutic efficacy in cancer treatment. This review tries to highlight the existing literature.
Major Comments

  1. The review highlights curcumin’s poor absorption and rapid metabolism, but does not offer detailed discussion on the most promising third-generation formulations and their comparative efficacy in clinical settings. Please expanding on recent advancements in delivery systems (e.g., nanoparticles, liposomes) to strengthen the translational relevance.
  2.  While curcumin enhances T-cell and NK cell activity, it may impair dendritic cell function, which could hinder antigen presentation. Authors should address this dichotomy in detail, discussing potential strategies to mitigate negative effects while stating immunostimulatory benefits.
  3. Much of the cited research relies on preclinical models (cell lines, mice). Please include more human clinical trial data—especially on curcumin’s synergy with immunotherapies—will support the case for its therapeutic potential.
  4. The proposed synergies with ICIs, ACT, and cancer vaccines are compelling but lack mechanistic depth. A clearer explanation of how curcumin remodels the TME to enhance these therapies would improve scientific depth.
  5. The manuscript does not adequately address optimal dosing, potential toxicity, or patient specific factors (e.g., cancer type, stage) which will influence curcumin’s efficacy. Adding a section on safety profiles and dose-response relationships could improve clinical applicability.

Minor Comments

  1. The figures (e.g., apoptosis pathways, macrophage polarization) are informative but could use simplified legends.
  2. Abbreviations like ‘TME’ (tumor microenvironment) are sometimes spelled out and other times abbreviated. Please keep uniform.

Remark

The review is informative and adequately organized. However, few concerns or improvements should be addressed.

Author Response

Major Comment #1

The review highlights curcumin’s poor absorption and rapid metabolism, but does not offer detailed discussion on the most promising third-generation formulations and their comparative efficacy in clinical settings. Please expanding on recent advancements in delivery systems (e.g., nanoparticles, liposomes) to strengthen the translational relevance.

Response #1

We agreed that there was limited discussion on the promising third-generation formulations and their comparative efficacy in clinical settings. We added a table 3 to boost the clinical significance and evidence in this paper, which includes recent clinical trials involving curcumin and various cancers. In this table there is multiple clinical trials cited that expands on these advancement systems. We thank you for pointing out this oversight and hope we addressed this adequately with the addition of clinical third generation data.

Major Comment #2

While curcumin enhances T-cell and NK cell activity, it may impair dendritic cell function, which could hinder antigen presentation. Authors should address this dichotomy in detail, discussing potential strategies to mitigate negative effects while stating immunostimulatory benefits.

Response #2

We agreed with this comment and decided that it needed to be fixed. Another reviewer recommended that reframe curcumin and DC in a cancer context by noting DCs can become tolerogenic and curcumin may prevent this. We also received other comments from reviewers that were like this one that noted multiple dichotomies that needed to be summarized or addressed and how to potentially address this clinically. To address these, we created a new section (Section 5. Contextual Dualities) which addresses the major dualities presented by curcumin use. We focused on the T-cell dichotomy, i.e. curcumin prevent naïve T-cell activation and differentiation, yet it also promotes persistence and infiltration, as we viewed this most clinically relevant. We suggested how ICIs, CAR-T, and vaccines may effected by this T-cell dichotomy and how the benefits could be maximized while the detriments could be minimized.

Major Comment #3

Much of the cited research relies on preclinical models (cell lines, mice). Please include more human clinical trial data—especially on curcumin’s synergy with immunotherapies—will support the case for its therapeutic potential.

Response #3

We agreed with this comment and included a new table (table 3) to address this. We included recent clinical trials and ongoing clinical trials that included curcumin and various cancers in different contexts. It is of note that there is a lack of clinical trials exploring immunotherapies in conjunction with curcumin. We also added another paragraph in section 6 (Challenges and Future Directions) to clarify and introduce what this table is. We believe this boosts our case for clinical applicability and safety in multiple cancer types and conditions.

Major Comment #4

The proposed synergies with ICIs, ACT, and cancer vaccines are compelling but lack mechanistic depth. A clearer explanation of how curcumin remodels the TME to enhance these therapies would improve scientific depth.

Response #4

We agreed that our discussion of immunotherapies in section 4 lacked depth. To address this, we included changes to every subsection of section 4. We focused our changes largely for ACT, we included three new paragraphs which presents how curcumin may mechanistically boost efficacy. Additionally, we added multiple new citations to this section. We also added new information to the cancer vaccine portion of this paper to improve depth.

Major Comment #5

The manuscript does not adequately address optimal dosing, potential toxicity, or patient specific factors (e.g., cancer type, stage) which will influence curcumin’s efficacy. Adding a section on safety profiles and dose-response relationships could improve clinical applicability.

Response #5

We agreed with this comment and thought that this should be included in our paper for increased clinical applicability as well as completeness on the topic. Given the need to organize this data and improve clinical applicability in the paper, we thought the best way to implement this information was in a table. Table 3 was created which included recent and ongoing clinical trials which included summaries of these trials which included information about safety, dosage, cancer type, and results when the data was available. We hope this adequately addresses this concern. 

Minor Comment #1

The figures (e.g., apoptosis pathways, macrophage polarization) are informative but could use simplified legends.

Minor Response #1

We agreed that the figure legends could be simplified and had unnecessary or redundant information. We shortened these legends to be more readable.

Minor Comment #2

Abbreviations like ‘TME’ (tumor microenvironment) are sometimes spelled out and other times abbreviated. Please keep uniform.

Minor Response #2

We agreed with this comment and went through the paper to address this issue. We believe we have fixed this.

Round 2

Reviewer 1 Report

Comments and Suggestions for Authors

Accept